# Schistosomiasis and soil-transmitted helminthiasis preventive chemotherapy: Adverse events in children from 2 to 15 years in Bengo province, Angola

**Manuel Lemos**[1,2,3], **Joao M. Pedro**[1,2]*, **Cláudia Fançony**[1,2], **Sofia Moura**[1], **Miguel Brito**[1,4], **Susana Vaz Nery**[5], **Carlos Pinto Sousa**[3], **Henrique Barros**[2,6]

**1** CISA—Centro de Investigação em Saúde de Angola, Caxito, Angola, **2** EPIUnit, Instituto de Saúde Pública, Universidade do Porto, Porto, Portugal, **3** Departamento de Saúde Pública, Faculdade de Medicina, Universidade Agostinho Neto, Luanda, Angola, **4** H&TRC—Health and Technology Research Center, Escola Superior de Tecnologia da Saúde de Lisboa, Instituto Politécnico de Lisboa, Lisboa, Portugal, **5** Kirby Institute, University of New South Wales, Sydney, Australia, **6** Faculdade de Medicina, Universidade do Porto, Porto, Portugal

* joao.almeidapedro@cisacaxito.org

**Data Availability Statement:** The anonymised data set is freely available in Zenodo database (DOI: 10.5281/zenodo.3660023).

## Abstract

Preventive chemotherapy campaigns with praziquantel and albendazole are being implemented in Angola, as a high priority public health intervention. However, there are no published data regarding adverse events associated with these medications. In this context, we analysed adverse events due to co-administration of praziquantel and albendazole in endemic areas of schistosomiasis and soil-transmitted helminths in Bengo, Angola. In the context of a targeted drug administration, between December 2012 and September 2013, we conducted two surveys after co-administrating single oral doses of praziquantel and albendazole tablets to children 2 to 15 years of age. About 24 hours after each treatment, participants answered a questionnaire about adverse events. At baseline, 605 children (55.0% male; mean age: 9.7 years) were treated; 460 were interviewed and 257 (55.9%) reported at least one adverse event, 62.3% (160/257) of children being infected with *schistosoma haematobium*. After six months of treatment, among 339 children surveyed, 184 (54.3%) reported adverse events, with 49.5% (91/184) of infected children. Adverse events were most common in preschool-aged children, with no significant difference between genders. The most frequent adverse events in the two surveys were abdominal pain (18.5%, 25.7%), headache (20.9%, 23.0%) and dizziness (15.7%, 19.8%). Children aged 12 to 15 years (adjusted OR = 0.40, *p* = 0.040) and those with mixed infection (adjusted OR = 0.04, *p* = 0.011) had lower odds of adverse events. After the second treatment, those with heavy infection (adjusted OR = 2.72, *p* = 0.018) and aged 9–11 years (adjusted OR = 2.01, *p* = 0.049) had significantly fewer adverse events. About 2.0% of children experienced severe adverse events. This study adds evidence that preventive chemotherapy for schistosomiasis and soil-transmitted helminths control is safe, but cases of adverse events are expected. Standardized methodologies to discriminate drug-related adverse events from the clinical manifestations of the infections are needed.

**Funding:** This work was supported by the promoters of the CISA as follows: Camões, Institute of Cooperation and Language, Portugal (www.instituto-camoes.pt/en/); Calouste Gulbenkian Foundation, Portugal (https://gulbenkian.pt/en/); Government of Bengo Province; Angolan Ministry of Health (www.minsa.gov.ao). The funders had no role in study design, data collection and analysis, decision to publish, or preparation of the manuscript.

## Introduction

Neglected tropical diseases (NTD), including schistosomiasis and soil-transmitted helminths (STHs), remain important public health issues in developing countries, where lack of adequate clean water and poor sanitation are common [1–3]. Schistosomiasis is one of the most socio-economically devastating parasitic disease in the world, causing estimated 280 thousand deaths and 4.5 million Disability Adjusted Life Years (DALYs) [4]. On the other hand, STHs, namely infections by *Ascaris lumbricoides*, *Trichuris trichiura* and hookworms, affect more than one billion individuals worldwide, causing respectively, 60, 10 and 65 thousand deaths, respectively [5,6]. In Angola, the prevalence of those diseases are high and variable within the country [7]. Recently, in a community based survey conducted by Figueiredo *et al* [8] in the Dande municipality (northern Angola), prevalence of urogenital schistosomiasis in preschool-aged children (PSAC) and school-aged children (SAC) were reported to be 10% and 17%, respectively, with 10% of anaemia cases in the studied population. The same study determined the prevalence of infection by at least one STH of 22.6% in preschool-aged children and 31.6% in school-aged children in the Dande municipality [8]. Another study conducted in the same region found an association between *A. lumbricoides* infection and malnutrition in children [9].

Preventive chemotherapy (PC) refers to the periodic administration of anthelminthic drugs to populations at risk of morbidity, with the aim to reduce infection intensity and eliminate moderate and heavy infections [10]. Thus, mass drug administration (MDA) is the corner-stone of the current global strategy to control helminthiasis. In children aged 2 to 15 years, Schistosomiasis is recommended to be controlled by taking a single dose (40mg/Kg child weight) of praziquantel (PZQ) and STHs with a single dose (400mg) of albendazole (ALB) [11]. ALB has a remarkable safety record, considering that a very low frequency and incidence of adverse events (AEs), mainly gastrointestinal, are described in the literature [12,13]. On the other side, despite that minimal and transient adverse events are described to occur with PZQ (reported mainly in heavily infected people and normally associated to the response of the host immune system to the dying worms), serious adverse reactions can also take place and may reduce drug compliance [13–21]. In fact, AEs resulting from co-administrating both drugs are described to be similar in severity to those experienced with monotherapy, however, information regarding the safety of this combination in children under 4 years of age (or under 94 cm in height) is scarce, especially in Angola [22]. As a consequence, preschool-aged children (PSAC) were considered ineligible for MDA with PZQ and the control approach were the referral to health facilities for individual case management [11].

According to WHO guidelines [10,11], AEs occur 24–48 hours after drug administration but do not necessarily have a causal relationship with treatment. They may be mild when not affecting daily activities (e.g. playing) or moderate when affecting the performance of daily activities. However, severe AEs require complete rest and / or medication, while serious AEs are life-threatening and require admission to the hospital.

In 2010 WHO recognized that preschool-aged children were a high-risk group for schisto-somiasis (when living in highly endemic settings), that should also be included in public health interventions, along with school-aged children and woman in childbearing age [15,23]. This was followed by several reports on the safety of PZQ in preschool-aged children (between 1 month and 7 years) and the recognition that administrating PZQ to those children was safe and efficacious [11,18,23–25]. Nevertheless, Angola is an endemic country for schistosomiasis and STHs and the therapeutic policy for those diseases is presently aligned with the 2006 WHO guidelines [11]. Currently, and at a national level, diagnosed cases of schistosomiasis in school-aged children are recommended to be medically treated at health care units and control activities for STHs are based in deworming campaigns [7,11].

Therefore, further evidence on adverse events (AEs) to the co-administration of PZQ and ALB, experienced in the context of TDA campaigns, are needed. Additionally, there is no published data in Angola regarding adverse events associated to the co-administration of PZQ and ALB, either in preschool-aged (particularly in under 4) or school-aged children. Thus, considering that the local health authorities of the NTD control program are implementing regular helminth control campaigns, these data are essential to inform these programs and to fill the treatment gap for children under 4 (currently excluded from TDA and often neglected).

Consequently, this pioneering study was carried out in the community of Cabungo and the Porto Quipiri School, located in the study area of the CISA project (Angola Health Research Centre, translated). We aimed to analyse the adverse events experienced following co-administration of PZQ and ALB for preventive chemotherapy against schistosomiasis and STHs.

## Methods

### Study site

We conducted this study between December 2012 and September 2013, in Cabungo community and Porto Quipiri School, located in the study area of CISA, in the Dande Municipality, Bengo Province—northern Angola. These hamlets were selected due to high prevalence of SCH, STHs and malaria observed in the area, whose structure, dynamics and geographical distribution of the population was previously described by Costa *et al* (2012) [26] and the epidemiology of schistosomiasis and STHs was detailed recently by Sousa-Figueiredo *et al* (2012) [8].

### Study design and aim

This is a longitudinal study, conducted in the context of a plan intervention by local health authorities, aiming to investigate the adverse events experienced by PSAC and SAC following the mass drug co-administration of PZQ and ALB, for the control of schistosomiasis and STHs at baseline and 6 months after the first treatment.

### Participants and therapeutic intervention

For this study, we enrolled 605 children, of whom 460 (76.0%) and 339 (56.0%) successfully participated in the first and second survey for AEs, respectively. All children, aged between 2 to 15 years old and living in those areas, were invited to participate and received a kit containing single doses of PZQ (40mg/kg), using the dose pole method to determinate the number of PZQ tablets to be administered to each child, and single doses of ALB (400mg) tablets. The tablets were crushed or broken when necessary and given with water or juice, as recommended by others [26], taken under direct observation of a health professional onsite. We also timely treated, with artemeter-lumefantrine (AL-20/120mg), children tested positive for malaria as recommended by the National Malaria Control Program [27]. In addition, refusals or failure in successful administration was documented. We excluded children if any severe adverse events were reported to be experienced in previous administration of PZQ and ALB.

### Laboratory analysis

Pre-treatment stool samples were collected for the diagnosis of intestinal parasites (*Schistosoma mansoni*, hookworms, *Ascaris lumbricoides*, *Trichuris trichuria* and *Hymenolepis nana*), performed by Kato-Katz method [28,29], and the presence and intensity of *S. haematobium* was determined by the examination of the pellet resulting from 10ml centrifuged pre-treatment urine [28]. The intensity of *S. haematobium* infection was recorded as light, moderate

and heavy if 1–49, 50–499 and equal or more than 500 eggs per 10 ml of urine were observed, respectively, according to WHO recommendations [30]. We collected capillary blood samples for the diagnosis of uncomplicated malaria, performed by rapid diagnostic tests (RDTs) according to the manufacturer (SD BIOLINE Malaria Ag P.f/P.v, Standard Diagnostics, Inc). For the measurement of haemoglobin levels, we used the Hemocue System (HemoCue® 201 +, Angelholm, Sweden).

## Adverse events data collection

We used a structured questionnaire to collect information on the adverse events (AEs) experienced by the children, performed to caretakers between 24 and 72 hours after treatment. When the caretaker was, absent and children had discernment to respond, the interview was performed with the children. Mild-to-moderate AEs were defined as undesirable experiences following drug administration, similarly to others, and severe AEs were defined as symptoms and signs due to administration drugs, which forced parents or caretaker to take their children to the health facility to be observed or hospitalized [31].

## Statistical analysis

The collected data was first entered into the CISA database and then statistical analysis was conducted with SPSS version 23.0 computer software (IBM Corporation, New York, USA). Pearson's chi-square tests were used to compare the occurrence of AEs between the two surveys and for categorical variables. Alternatively, Fisher's exact test was used when any 2x2 contingency table cell expected a count below five.

We performed binary logistic model to identify possible independent predictors of AEs in the two surveys, using adjusted odds ratios (OR) and their 95% confidence interval (CI). The threshold for significant level was 0.05.

## Ethics

All procedures performed in this study were in accordance with the standards of the 1964 Declaration of Helsinki and its later amendments. The Ethics Committee of the Angolan Ministry of Health approved the study protocol and all use of secondary data. Written informed consent was obtained and signed by a guardian, parent or a caretaker for each of the participants. A copy of the signed consent form, as well as contact information, was subsequently delivered to each participant. The Provincial health office, local leaders and parents were previously informed about the study in the area.

## Results

A total of 605 children aged between 2 to 15 years (55.0% of males, 333/605) mean age 9.7 years ± 3.5, were subject to preventive chemotherapy with PZQ and ALB. Children with RDT positive for malaria were also treated with AL. At the baseline, of the 605 participating children, 548 took PZQ + ALB tablets while 57 took PZQ + ALB + AL. From those, 460 interviews were performed (76.0%, 460/605), responded mainly by the children themselves (75.0%) and guardians (25.0%). AEs were reported by 257 (55.9%, 257/460) children, 22 PSAC and 235 SAC. In this group of participants, 62.3% (160/257) were infected with either *S. haematobium* (100.0%, 160/160), STHs (20.6%, 33/160) and/or *Plasmodium falciparum* (19.4%, 31/160). On the other hand, 64.0% (130/203) of children, that had not experienced AEs, were also infected with those parasites (Fig 1). At the sixth month follow-up, 339 children included at the baseline, were surveyed for AEs after second round of medications (with PZQ+ALB or with PZQ

+ALB+AL), from which 184 (54.3%, 184/339) reported to have experienced at least one adverse event, 19.6% (36/184) PSAC and 80.4% (148/184) SAC, as shown in Fig 1.

No significant difference in the characteristics between participants and dropout children were observed at baseline. However, after the second round of treatment, a significantly lower proportion of PSAC (98.1% to 1.9%, $p < 0.001$) and SAC (75.9% to 24.1%, $p < 0.001$) were observed in the dropout group (Table 1).

In the first survey, headache (20.9%, 96/460), abdominal pain (18.5%, 85/460) and dizziness (15.7%, 72/460) were the three most frequently reported AEs. PSAC experienced mainly headache (18.6%), vomiting (14.3%) and abdominal pain (7.1%), whereas SAC, after headache (21.3%) and abdominal pain (20.5%) reported dizziness (18.2%). PSAC experienced significantly more vomiting than SAC (14.3% vs 6.7%, $p = 0.029$). SAC in turn experienced more abdominal pain (20.5% vs 7.1%, $p = 0.008$) and dizziness (18.2% vs 1.4%, $p < 0.001$). Blood in stool and light sensitivity was only reported for SAC. After the second treatment, reports of abdominal pain, headache, dizziness and fatigue were also more frequent.

In this survey, PSAC reported significantly more fatigue (30.2% vs 15.4%, $p = 0.009$), blood in urine (20.8% *vs* 5.6%, $p < 0.001$) and blood in stool (7.5% *vs* 1.0%, $p = 0.002$). Light sensitivity was not experienced by PSAC in both treatments, neither breathing difficulty after the second treatment (see Table 2).

Although, 2.3% (6/257) of children reporting AEs after the first treatment, and 1.6% (3/184) after the second treatment searched for medical attention in a nearby health facility but no admission notification was reported. Data on hospital care was not collected and therefore classification of the seriousness of AEs was not possible to perform.

Most children reported two simultaneous AEs followed by those reporting only one event (32.1% *vs* 31.5% in the first survey and 30.7% *vs* 29.2% in the second survey). There were also children who reported three or more simultaneous events. The most common single AEs were abdominal pain (37.9%, 22/58), headache (29.3%, 17/58) and fatigue (17.2%, 10/58) while double AEs were "abdominal pain + headache" (15.3%, 9/59), "fatigue + dizziness" (11.9%, 7/59) and "abdominal pain + fatigue" (6.8%, 4/59). Triple AEs were "abdominal pain + headache + dizziness" (12.1%, 4/33) and "abdominal pain + fatigue + dizziness" (6.1%, 2/33).

In the first survey, we observed that children with mixed infections had lower odds ratio to develop AEs than those with simple infections (adjusted OR = 0.14, p = 0.016 vs adjusted OR = 0.04, p = 0.011). Only the age group from 12 to 15 years old presented a significant OR of developing AEs (adjusted OR = 0.40, p = 0.040). All of these associations observed in the first survey lost their statistical significance after the second treatment. However, children with severe S. haematobium infection (adjusted OR = 2.72, p = 0.018) and children aged 9 to 11 years (adjusted OR = 2.01, p = 0.049) were significantly more probability to have AEs. (Table 3).

## Discussion

In this study, 55.9% of the children who completed both surveys reported having experienced AEs after the first treatment with ALB, PZQ and/or AL and 54.3% after the second treatment. In general, the main AEs reported were abdominal pain, headache and dizziness, both after the first and after the second treatment, and only a low proportion of children searched for medical attention after both treatments (2.3% and 1.6%, respectively). The present study reported different proportions of AEs comparable to those found in other studies involving praziquantel and / or albendazole. However, there are variations that may occur due to the heterogeneous antecedents of the participating individuals, such as age, nutritional and immunological status, socioeconomic conditions, environmental exposure, prevalence and intensity of

```
┌─────────────────────────────────────────────────────────────────────┐
│                              Enrolled                                 │
│                            630 children                  Dropouts     │
│                            (2-15 years)                      ↓        │
└─────────────────────────────────────────────────────────────────────┘

                                    25 excluded – missed PZQ and ALB:
                                    8-cried, 7-vomited, 5-spit,
                                    4-refused and 1-breast-feeding.

┌─────────────────────────────────────────────────────────────────────┐
│                        First medication round                         │
│                            605 children                               │
│          (548 medicated with PZQ+ALB and 57 medicated with PZQ+ALB+AL) │
└─────────────────────────────────────────────────────────────────────┘

                                    145 absents in survey day.

┌─────────────────────────────────────────────────────────────────────┐
│                         First survey for AEs                          │
│                            460 children                               │
│            (70 preschool and 390 school-aged children)                │
├──────────────────────────────────┬────────────────────────────────────┤
│  257 experienced at least one AE  │       203 without AEs              │
│  22 PSAC (9 male; 13 infected)    │  48 PSAC (30 male; 29 infected)    │
│  235 SAC (124 male; 147 infected) │  155 SAC (95 male; 101 infected)   │
│           160 infected            │         130 infected               │
│ SCH (100%), STHs (20.6%) and      │ SCH (100%), STHs (23.1%) and       │
│        Malaria (19.4%)            │        Malaria (14.6%)             │
└──────────────────────────────────┴────────────────────────────────────┘

                                    29 follow-up losses.

┌─────────────────────────────────────────────────────────────────────┐
│                       Second medication round                         │
│                            431 children                               │
│       (391 medicated with of PZQ+ALB and 40 medicated with PZQ+ALB+AL) │
└─────────────────────────────────────────────────────────────────────┘

                                    92 excluded:
                                    84-absents in survey day,
                                    7-no response and 1-vomited.

┌─────────────────────────────────────────────────────────────────────┐
│                        Second survey for AEs                          │
│                            339 children                               │
│            (53 preschool and 286 school-aged children)                │
├──────────────────────────────────┬────────────────────────────────────┤
│  184 experienced at least one AE  │       155 without AEs              │
│  36 PSAC (27 male; 16 infected)   │  17 PSAC (4 male; 5 infected)      │
│  148 SAC (83 male; 75 infected)   │  138 SAC (79 male; 63 infected)    │
│           91 infected             │         68 infected                │
│ SCH (89.6%), STHs (16.0%) and     │ SCH (85.3%), STHs (16.0%) and      │
│        Malaria (0.03%)            │        Malaria (5.3%)              │
└──────────────────────────────────┴────────────────────────────────────┘
```

**Fig 1. Study participants and dropouts.** PSAC, preschool-aged children; SAC, school-aged children; SCH, schistosomiasis; SHTs, soil-transmitted helminths; AEs, adverse events; PZQ, praziquantel; ALB, albendazole; AL, Artemeter-Lumefantrine.

infection, stage of parasite development, etc[13,18,20,21,31–37]. Zwang *et al* reviewed the efficacy and safety of PZQ (40mg/kg), and report that the main AEs experienced and their incidences are mainly abdominal pain (31.8%), muscle pain (29.2%), joint pain (20.2%), headache (13.6%), diarrhoea (12.9%), fatigue (9.6%), nausea (10.6%), dizziness (11.9%), vomiting (7.9%) and itching (9.8%) [18]. Additionally for ALB, there are reports of the occurrence of epigastric pain, dry mouth, fever and itching [18,21,32,38–40]. However, the occurrence of AEs are reported to be associated with the proportion of dying S. *haematobium* worms, i.e., with the pharmacologic effect of the drug on the parasite (for example; abdominal pain is reported to be associated with the deposition of dead worms in the mesenteric veins), they can also occur due to the natural course of disease [19,32,37,41,42].

Neumayr *et al* mentions that symptomatic acute schistosomiasis or treatment-induced reactions can manifest themselves with identical symptoms [32]. This group and others further discuss that the exposure to a high level of parasite antigens, caused by larval migration and maturation (of helminth larvae) or early oviposition in symptomatic acute schistosomiasis, can lead to an immune overreaction, resulting in symptoms that are similar to those reported as AEs to PZQ, but totally independent from treatment [19,32,43–45]. Although increasing the number of infections significantly reduced the chances of AE occurrence, we found that heavy S. haematobium infection was significantly associated with AEs occurrence, with odds ratio twice as high in individuals with this parasitic load in the second survey.Also, *H. nana* infections alone can cause headache, dizziness, abdominal pain and diarrhoea, *A. lumbricoides* can cause nausea, vomiting, diarrhoea and abdominal pain, *T. trichiura* can cause abdominal pain and diarrhoea, and *S. stercoralis* can cause erythema, itching, fever, abdominal pain and

**Table 1. Characteristics of participants and dropouts in AEs survey.**

| Characteristics | Baseline (n = 605) | | | 6 months follow-up (n = 431) | | |
|---|---|---|---|---|---|---|
| | Participants n = 460 (76.0%) | Excluded n = 145 (24.0%) | *P value** | Participants n = 339 (78.7%) | Excluded n = 92 (21.3%) | *P value** |
| **Age group** | | | | | | |
| **PSAC**[a] | 70 (77.8) | 20 (22.2) | 0.674 | 53 (98.1) | 1 (1.9) | <0.001 |
| **SAC**[b] | 390 (75.7) | 125 (24.3) | | 286 (75.9) | 91 (24.1) | |
| **Gender** | | | | | | |
| **Male** | 258 (77.5) | 75 (22.5) | 0.357 | 193 (80.1) | 48 (19.9) | 0.415 |
| **Female** | 202 (74.3) | 70 (25.7) | | 146 (76.8) | 44 (23.2) | |
| ***S. haematobium* eggs** | | | | | | |
| **Positive** | 290 (76.1) | 91 (23.9) | 0.951 | 159 (81.5) | 36 (18.5) | 0.184 |
| **Negative** | 170 (75.9) | 54 (24.1) | | 180 (76.3) | 56 (23.7) | |
| **Soil-transmitted helminths eggs** | | | | | | |
| **Positive** | 156 (80.4) | 38 (19.6) | 0.058 | 58 (90.6) | 6 (9.4) | 0.603 |
| **Negative** | 205 (71.7) | 81 (28.3) | | 163 (87.2) | 24 (12.8) | |
| **Missed examination** | 99 (79.2) | 26 (20.8) | | 118 (65.6) | 62 (34.4) | |

* chi-square test

[a] Preschool-aged children

[b] School-aged children.

**Table 2. Frequency of reported AEs of children treated with PZQ and ALB in two medication rounds.**

| Adverse events | 1st survey | | | | 2nd survey | | | |
|---|---|---|---|---|---|---|---|---|
| | PSAC[a] 70 (15%) | SAC[b] 390 (85%) | Total 460 (100%) | p value* | PSAC[a] 53 (16%) | SAC[b] 286 (84%) | Total 339 (100%) | P value* |
| **Overall** | **22 (31.4)** | **235 (60.3)** | **257 (55.9)** | **<0.001** | **36 (67.9)** | **148 (51.7)** | **184 (54.3)** | **0.043** |
| Abdominal pain | 5 (7.1) | 80 (20.5) | 85 (18.5) | 0.008 | 19 (35.8) | 68 (23.8) | 87 (25.7) | 0.065 |
| Headache | 13 (18.6) | 83 (21.3) | 96 (20.9) | 0.607 | 13 (24.5) | 65 (22.7) | 78 (23.0) | 0.775 |
| Dizziness | 1 (1.4) | 71 (18.2) | 72 (15.7) | <0.001 | 9 (17.0) | 58 (20.3) | 67 (19.8) | 0.580 |
| Fatigue | 1 (1.4) | 29 (7.4) | 30 (6.5) | 0.061 | 16 (30.2) | 44 (15.4) | 60 (17.7) | 0.009 |
| Vomiting | 10 (14.3) | 26 (6.7) | 36 (7.8) | 0.029 | 2 (3.8) | 33 (11.5) | 35 (10.3) | 0.088 |
| Blood in urine | 1 (1.4) | 29 (7.4) | 30 (6.5) | 0.061 | 11 (20.8) | 16 (5.6) | 27 (8.0) | <0.001 |
| Fever | 1 (1.4) | 27 (6.9) | 28 (6.1) | 0.077 | 2 (3.8) | 21 (7.3) | 23 (6.8) | 0.343 |
| Itching | 2 (2.9) | 30 (7.7) | 32 (7.0) | 0.143 | 4 (7.5) | 13 (4.5) | 17 (5.0) | 0.358 |
| Diarrhoea | 2 (2.9) | 18 (4.6) | 20 (4.3) | 0.507 | 3 (5.7) | 11 (3.8) | 14 (4.1) | 0.542 |
| Light sensitivity | 0 (0.0) | 25 (6.4) | 25 (5.4) | 0.029 | 0 (0.0) | 7 (2.4) | 7 (2.1) | 0.250 |
| Joint pain | 2 (2.9) | 15 (3.8) | 17 (3.7) | 0.686 | 1 (1.9) | 6 (2.1) | 7 (2.1) | 0.921 |
| Blood in stool | 0 (0.0) | 4 (1.0) | 4 (0.9) | 0.395 | 4 (7.5) | 3 (1.0) | 7 (2.1) | 0.002 |
| Red skin | 1 (1.4) | 8 (2.1) | 9 (2.0) | 0.729 | 1 (1.9) | 3 (1.0) | 4 (1.2) | 0.604 |
| Breathing difficulty | 2 (2.9) | 11 (2.8) | 13 2.8) | 0.061 | 0 (0.0) | 3 (1.0) | 3 (0.9) | 0.454 |

* chi-square test

[a] Preschool-aged children

[b] School-aged children.

diarrhoea [35,38,46]. Thus, we postulate that children with mixed infections between schistosomiasis, STH and hymenolepiasis can be manifesting additive drug-unrelated AEs, overestimating the frequency of AEs reported here.

Despite that other authors have found no progression of AEs between treatment rounds, we found that after the second treatment the reports of abdominal pain, headache, dizziness and fatigue generally increased and the reports of joint pain and breathing difficulty generally decreased [20]. Additionally to that, experiencing 2 simultaneous AEs was slightly more frequent than experiencing only one event, and experiencing 3, 4 or 5 simultaneous events was also reported, despite of less frequent. Considering that the literature reports an association between AEs and the intensity of infection and anemia status, we found a statistically significant association between heavy infection and AEs [18,20,36]. Nevertheless, it should be considered that the administration of PZQ following a high-lipid or high-carbohydrate diets can increase its bioavailability (by a factor of 2.7 and 3.9) and in turn may alter the pharmacokinetic profile of the drug, possibly influencing the effect of the drug on the parasite and probably the frequency and type of AEs [18,19,33,41]. In this study children aged 9 to 15 years old, were at lower risk of experiencing AEs than younger children, however, it should be considered that this association lost their statistical significance after the second treatment and that the number of individuals within the PSAC group is small. Thus, these results should be interpreted carefully. Nevertheless, similar frequencies of AEs were observed by others between preschool and school children [21].

Regarding the 45% of "uninfected" children experiencing AEs in our study, in addition to the fact that these children may have AEs due to drug use only; we also considered the stages of infection undetected by the diagnostic technique used here [43,47]. The limited effect of PZQ on young forms of parasites and the progression of disease would still manifest signs and symptoms [40,43,47].

**Table 3. Association between AEs and characteristics of participants in two AEs survey moments.**

| Characteristics | 1st survey | | | 2nd survey | | |
|---|---|---|---|---|---|---|
| | Total 257/339 (75.8%) | Adjusted 0R (95% CI) | *P value* | Total 184/339 (54.3%) | Adjusted OR (95% CI) | *P value* |
| **Gender** | | | | | | |
| Female | 113/146 (77.4) | 1 | - | 75/146 (51.4) | 1 | - |
| Male | 144/193 (74.6) | 0.96 (0.54–1.70) | 0.904 | 109/193 (56.5) | 1.18 (0.75–1.84) | 0.463 |
| **Age group** | | | | | | |
| 2–5 y | 45/55 (81.8) | 1 | | 25/55 (45.5) | 1 | - |
| 6–8 y | 51/65 (78.5) | 0.80 (0.30–2.09) | 0.652 | 38/65 (58.5) | 1.95 (0.91–4.18) | 0.085 |
| 9–11 y | 84/111 (75.7) | 0.66 (0.28–1.59) | 0.366 | 64/111 (57.7) | 2.01 (1.00–4.05) | 0.049 |
| 12–15 y | 77/108 (71.3) | 0.40 (0.16–0.95) | 0.040 | 57/108 (52.8) | 1.69 (0.84–3.39) | 0.139 |
| ***S. haematobium* eggs** | | | | | | |
| Negative | 97/130 (74.6) | 1 | - | 102/195 (52.3) | 1 | - |
| Light (1-49eggs) | 17/25 (68.0) | 0.63 (0.23–1.71) | 0.370 | 28/46 (60.9) | 1.43 (0.73–2.80) | 0.291 |
| Moderate (50-499eggs) | 75/100 (75.0) | 1.13 (0.59–2.19) | 0.700 | 29/62 (46.8) | 0.73 (0.40–1.35) | 0.324 |
| Heavy ($\geq$500eggs) | 68/84 (81.0) | 2.11 (0.76–3.26) | 0.213 | 25/36 (69.4) | 2.72 (1.18–6.24) | 0.018 |
| **STHs Eggs** | | | | | | |
| Negative | 205/251 (81.7) | 1 | - | 159/298 (53.4) | 1 | - |
| Positive | 52/88 (59.1) | 2.11 (0.42–10.57) | 0.361 | 25/41 (61.0) | 0.43 (0.08–2.15) | 0.310 |
| ***Hymenolepis nana* eggs** | | | | | | |
| Negative | 241/313 (77.0) | 1 | - | 177/328 (54.0) | 1 | - |
| Positive | 16/26 (61.5) | 1.77 (0.36–8.51) | 0.475 | 7/11 (63.6) | 1.95 (0.41–9.11) | 0.395 |
| ***Plasmodium* parasites** | | | | | | |
| Negative | 235/307 (76.5) | 1 | - | 179/327 (54.7) | 1 | - |
| Positive | 22/32 (68.8) | 0.82 (0.34–1.94) | 0.653 | 5/12 (41.7) | 0.59 (0.17–2.00) | 0.402 |
| **Number of infections** | | | | | | |
| Null | 191/228 (83.8) | 1 | - | 159/298 (53.4) | 1 | - |
| One | 59/94 (62.8) | 0.14 (0.30–0.69) | 0.016 | 21/33 (63.6) | 3.02 (0.58–15.69) | 0.188 |
| Two | 7/17 (41.2) | 0.04 (0.01–0.49) | 0.011 | 4/8 (50.0) | 1.67 (0.50–5.63) | 0.402 |
| **Anaemia** | | | | | | |
| No | 65/82 (79.3) | 1 | - | 89/159 (56.0) | 1 | - |
| Yes | 192/257 (74.7) | 0.69 (0.36–1.34) | 0.282 | 95/180 (52.8) | 0.78 (0.50–1.24) | 0.309 |

CI: confidence interval, OR: odds ratio, SHT: soil-transmitted helminths.

## Conclusions

This study adds evidence that chemotherapy for the control of schistosomiasis and soil-transmitted helminthiasis with PZQ and ALB is safe, although mild to moderate cases of AEs are expected. In these two surveys of AEs, we recorded 3.1% of children reporting medical attention due AEs after the medication. To assess the safety of co-administration of PZQ and ALB with artemeter-lumefantrine for positive cases of uncomplicated malaria in children, more standardized methodologies are needed to discriminate drug-related AEs from the clinical manifestations of the infections studied.

## Supporting information

**S1 File.**
(DOCX)

**S2 File.**
(PDF)

## Acknowledgments

We thank Calouste Gulbenkian Foundation for the project support and Health Research Centre of Angola (CISA, translated) for the technical assistance. We must not forget the valuable cooperation of the Provincial Health Department of Bengo, the National Program of Neglected Tropical Diseases and the Research Unit of the Porto Public Health Institute, in this project.

## Author Contributions

**Conceptualization:** Manuel Lemos, Joao M. Pedro, Cláudia Fançony, Miguel Brito, Susana Vaz Nery, Carlos Pinto Sousa, Henrique Barros.

**Data curation:** Manuel Lemos, Sofia Moura.

**Formal analysis:** Manuel Lemos, Joao M. Pedro.

**Funding acquisition:** Manuel Lemos, Miguel Brito.

**Investigation:** Manuel Lemos, Cláudia Fançony, Sofia Moura, Miguel Brito.

**Methodology:** Manuel Lemos, Joao M. Pedro, Henrique Barros.

**Project administration:** Manuel Lemos, Miguel Brito, Susana Vaz Nery, Carlos Pinto Sousa, Henrique Barros.

**Resources:** Manuel Lemos, Sofia Moura, Miguel Brito.

**Software:** Manuel Lemos.

**Supervision:** Miguel Brito, Henrique Barros.

**Validation:** Miguel Brito, Susana Vaz Nery, Carlos Pinto Sousa, Henrique Barros.

**Visualization:** Susana Vaz Nery.

**Writing – original draft:** Manuel Lemos.

**Writing – review & editing:** Manuel Lemos, Joao M. Pedro, Cláudia Fançony, Sofia Moura, Miguel Brito, Susana Vaz Nery, Carlos Pinto Sousa, Henrique Barros.

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
