## [Decision Letter · Decision Letter 0]

12 Dec 2019

PONE-D-19-22164

Schistosomiasis and soil-transmitted helminthiasis preventive chemotherapy: adverse events in children from 2 to 15 years in Bengo province, Angola

PLOS ONE

Dear Joao M Pedro,

Thank you for submitting your manuscript to PLOS ONE. After careful consideration, we feel that it has merit but does not fully meet PLOS ONE’s publication criteria as it currently stands. Therefore, we invite you to submit a revised version of the manuscript that addresses the points raised during the review process.

We would appreciate receiving your revised manuscript by 11th January 2020. To enhance the reproducibility of your results, we recommend that if applicable you deposit your laboratory protocols in protocols.io, where a protocol can be assigned its own identifier (DOI) such that it can be cited independently in the future. For instructions see: http://journals.plos.org/plosone/s/submission-guidelines#loc-laboratory-protocols

We look forward to receiving your revised manuscript.

Kind regards,

Mary Hamer Hodges

Academic Editor

PLOS ONE

Journal Requirements:

Additional Editor Comments (if provided):

The topic is of importance but as described by 2 reviewers the content needs review and the all three agree the overall presentation need improvement.

Reviewers' comments:

Reviewer's Responses to Questions

**Comments to the Author**

1. Is the manuscript technically sound, and do the data support the conclusions?

Reviewer #1: Yes

Reviewer #2: Yes

Reviewer #3: Yes

2. Has the statistical analysis been performed appropriately and rigorously? 

Reviewer #1: Yes

Reviewer #2: Yes

Reviewer #3: Yes

3. Have the authors made all data underlying the findings in their manuscript fully available?

Reviewer #1: Yes

Reviewer #2: Yes

Reviewer #3: Yes

4. Is the manuscript presented in an intelligible fashion and written in standard English?

Reviewer #1: Yes

Reviewer #2: Yes

Reviewer #3: Yes

5. Review Comments to the Author

Reviewer #1: This paper describes and quantifying the adverse affects of Praziquantel treatment in a cohort of people. The findings are important in raising awareness of adverse affects and also to show the level of severity or not. There are very few corrections needed and I have no specific comments. The English could be improved in places so I suggest the authors have a careful read to try and improve the English as needed.

Reviewer #2: Schistosomiasis and STH PC: Adverse events in children in Bengo province, Angola.

This ms is fine and adds to the information on adverse events during MDA and control programs in preschool age children and school age children. It looks at PZQ and ALB and concludes that treatment programs should continue despite small amounts of adverse events.

Title: fine

Abstract: fine – minor corrections –

L30 maybe put gender in here.

L32 correct spelling - Schistosoma from schistosome

L32 ‘follow-up’ – treatment or questionnaire – clarify

Introduction:

General points – if using acronyms spell out in first use then use the acronym

A clear definition of adverse events is needed; define the difference between mild/severe.

L56 -- ‘60, 10 and 65 thousand’ what? Unsure what is meant DALYs? Deaths?

L58 reference number needed for Figueiredo et al

L62 using geohelminth here – define.

L65 can used PC common acronym for preventative chemotherapy

L67 may be use mass drug administration (MDA) instead of targeted unless you are performing test and treat control strategy?

L70 use a ] instead of ) to enclose citation.

L74 consider word change to ‘normally’ from ‘possibly’

Methods:

Map?

L112 odd phrasing – ‘selected by convenience’?

L121 odd phrasing – ‘two deworming moments’ ?

L162 reference for the software

Ethical approval – is there an associated number for this study from the ethical committee?

It might be good to have the number of children enrolled in the study in your methods?

Results:

All tables – change ‘yes’ ‘no’ to egg positive negative or something. These need to be made clearer – particularly table 2 – it contains a lot of information but perhaps consider a graph or different way to display which highlights important results.

L185 change ‘themself’ to themselves and ‘caretakers’ to guardians

L188 Plasmodium in full not just P.

Discussion:

L259 – 262 Reword and break up – strange wording here.

L269 italicize S. haematobium

L273 ref in list please

L280 please clarify – not sure what it means

References:

Check formatting with the journal

Reviewer #3: This is an interesting report on adverse events during systematic preventive administration of praziquantel and albendazole in children. The prevalence in this area in Angola is impressive and thus the study area is adapted to have excellent data on the subject.

The paper is well written although the methodology and result section lack on clarity and can not be clear enough without the reading of the complementary documents : this should be improved.

There are some sentences that are unclear because of english or missing items : line 53 to 56 ; line 63 ; 70; 233 ; line 243-246.

In the discussion again it is not clear why the 45 % of uninfected children who have AES should be underdiagnosed ? why it is not possible to have AES with the drug alone ? (line 308)

These different points should be improved.

6. PLOS authors have the option to publish the peer review history of their article (what does this mean?). If published, this will include your full peer review and any attached files.

Reviewer #1: No

Reviewer #2: No

Reviewer #3: Yes: BISSER SYLVIE

---

## [Author Response · Author response to Decision Letter 0]

10 Jan 2020

We wish to thank this opportunity, careful revision and adequate comments that improve the final manuscript, as you may see on the enclosed documents. Further detailed answers to the reviewers are stated in the attached document.

Regarding the journal requirements, a full revision was made and we hope that everything is correct. In relation to data availability, we confirm that there is no ethical or legal restriction on sharing a de-identified data set that we will share upon acceptation of this manuscript.

---

## [Decision Letter · Decision Letter 1]

4 Feb 2020

Schistosomiasis and soil-transmitted helminthiasis preventive chemotherapy: adverse events in children from 2 to 15 years in Bengo province, Angola

PONE-D-19-22164R1

Dear Dr. Joao M Pedro,

We are pleased to inform you that your manuscript has been judged scientifically suitable for publication and will be formally accepted for publication once it complies with all outstanding technical requirements.

With kind regards,

Mary Hamer Hodges

Academic Editor

PLOS ONE

Additional Editor Comments (optional):

This article will be welcomed by programmers in highly endemic communities and enable the planning of medications to address the usually mild AEs experienced. It will also be useful to help inform parents, teachers and communities about AEs that are likely to occur and how best to prepare for them and manage them at large scale.

Reviewers' comments:

Reviewer's Responses to Questions

**Comments to the Author**

1. If the authors have adequately addressed your comments raised in a previous round of review and you feel that this manuscript is now acceptable for publication, you may indicate that here to bypass the “Comments to the Author” section, enter your conflict of interest statement in the “Confidential to Editor” section, and submit your "Accept" recommendation.

Reviewer #1: All comments have been addressed

Reviewer #3: All comments have been addressed

2. Is the manuscript technically sound, and do the data support the conclusions?

Reviewer #1: Yes

Reviewer #3: Yes

3. Has the statistical analysis been performed appropriately and rigorously? 

Reviewer #1: Yes

Reviewer #3: Yes

4. Have the authors made all data underlying the findings in their manuscript fully available?

Reviewer #1: Yes

Reviewer #3: Yes

5. Is the manuscript presented in an intelligible fashion and written in standard English?

Reviewer #1: Yes

Reviewer #3: Yes

6. Review Comments to the Author

Reviewer #1: (No Response)

Reviewer #3: (No Response)

7. PLOS authors have the option to publish the peer review history of their article (what does this mean?). If published, this will include your full peer review and any attached files.

Reviewer #1: No

Reviewer #3: No

---

## [Editor Report · Acceptance letter]

13 Feb 2020

PONE-D-19-22164R1 

Schistosomiasis and soil-transmitted helminthiasis preventive chemotherapy: adverse events in children from 2 to 15 years in Bengo province, Angola 

Dear Dr. Pedro:

I am pleased to inform you that your manuscript has been deemed suitable for publication in PLOS ONE. Congratulations! Your manuscript is now with our production department. 

With kind regards,

on behalf of

Dr. Mary Hamer Hodges 

Academic Editor

PLOS ONE